# Performance of Six Clinical Physiological Scoring Systems in Predicting In-Hospital Mortality in Elderly and Very Elderly Patients with Acute Upper Gastrointestinal Bleeding in Emergency Department

**DOI:** 10.3390/medicina59030556

**Published:** 2023-03-11

**Authors:** Po-Han Wu, Shang-Kai Hung, Chien-An Ko, Chia-Peng Chang, Cheng-Ting Hsiao, Jui-Yuan Chung, Hao-Wei Kou, Wan-Hsuan Chen, Chiao-Hsuan Hsieh, Kai-Hsiang Ku, Kai-Hsiang Wu

**Affiliations:** 1Department of Emergency Medicine, Chiayi Chang Gung Memorial Hospital, Chiayi County 613, Taiwan; 2Department of Emergency Medicine, Linkou Chang Gung Memorial Hospital, Taoyuan City 333, Taiwan; 3Department of Otorhinolaryngology-Head and Neck Surgery, Chiayi Chang Gung Memorial Hospital, Chiayi County 613, Taiwan; 4Department of Medicine, Chang Gung University, No. 259, Wenhua 1st Rd., Guishan Dist., Taoyuan City 333, Taiwan; 5Department of Emergency Medicine, Cathay General Hospital, Taipei City 106, Taiwan; 6Division of General Surgery, Department of Surgery, Linkou Chang Gung Memorial Hospital, Taoyuan City 333, Taiwan; 7Department of Pediatric, Chiayi Chang Gung Memorial Hospital, Chiayi County 613, Taiwan; 8Department of Emergency Medicine, Sijhih Cathay General Hospital, New Taipei City 221, Taiwan

**Keywords:** upper gastrointestinal bleeding, elderly patients, emergency department, scoring systems

## Abstract

*Background and Objectives:* The aim of this study is to compare the performance of six clinical physiological-based scores, including the pre-endoscopy Rockall score, shock index (SI), age shock index (age SI), Rapid Acute Physiology Score (RAPS), Rapid Emergency Medicine Score (REMS), and Modified Early Warning Score (MEWS), in predicting in-hospital mortality in elderly and very elderly patients in the emergency department (ED) with acute upper gastrointestinal bleeding (AUGIB). *Materials and Methods:* Patients older than 65 years who visited the ED with a clinical diagnosis of AUGIB were enrolled prospectively from July 2016 to July 2021. The six scores were calculated and compared with in-hospital mortality. *Results:* A total of 336 patients were recruited, of whom 40 died. There is a significant difference between the patients in the mortality group and survival group in terms of the six scoring systems. MEWS had the highest area under the curve (AUC) value (0.82). A subgroup analysis was performed for a total of 180 very elderly patients (i.e., older than 75 years), of whom 27 died. MEWS also had the best predictive performance in this subgroup (AUC, 0.82). *Conclusions:* This simple, rapid, and obtainable-by-the-bed parameter could assist emergency physicians in risk stratification and decision making for this vulnerable group.

## 1. Introduction

Acute upper gastrointestinal bleeding (AUGIB) is a common reason for emergency department (ED) visits and has high rates of morbidity and mortality [1,2]. In the United States, it accounted for >1 million ED visits and >780,000 hospital admissions in 2018 [3]. The mortality rate varied by region with an estimated range from 2% to 10% [4]. Studies conducted on Asian populations also revealed a similar mortality rate range [5,6]. Despite a decreasing trend in overall mortality and hospitalization rates over the previous decades due to advancements in medical and endoscopic management [7], AUGIB persistently has a much higher—and unacceptable—mortality rate in the geriatric population [8]. Therefore, elderly patients with AUGIB are regarded as a vulnerable group in need of further consideration [9].

In 2020, 727 million persons were 65 years or older, and the number is estimated to double to 1.5 billion in 2050, which is 16% of the total population [10]. An increase in the number of elderly people leads to increased demands for emergency department utility [11]. Elderly patients have higher comorbidities, vague clinical presentations, and polypharmacy status, which makes universal protocol-driven approaches unsuitable. Tailor-made approaches and management for specific diseases in elderly patients in the ED have been proven to be effective and benefit the prognosis. Although algorithms for major trauma, coronary artery disease, and cerebrovascular disease have been proposed, a customized method for managing elderly patients with AUGIB in the ED is lacking [12,13,14].

The cornerstone of AUGIB management includes rapid and precise risk stratification, adequate resuscitation and medical treatment, endoscopic hemostasis, and rescued radiologic or surgical intervention [4]. A recent consensus has emerged that the use of physiological scoring systems for risk stratification would help ensure consistent assessment and communication [15]. Scoring-system-based urgent endoscopy has also been proven to shorten the hospitalization period [16]. Although several physiological scoring systems have been proposed as predictors of mortality due to AUGIB in the ED, most have comprised advanced parameters such as endoscopic diagnosis and laboratory information. However, previous publications have proven that simple and obtainable-by-the-bed clinical physiological scoring systems are beneficial for risk stratification [17,18], but the performance of these rapid, easily calculated clinical scoring systems has not been studied in elderly populations. The aim of this study is to evaluate clinical physiological scoring systems for predicting in-hospital mortality in elderly patients with AUGIB in the ED.

## 2. Materials and Methods

### 2.1. Data Source and Study Population

This retrospective study was conducted from 1 July 2016, to 31 July 2021, at the emergency department of a university-affiliated teaching hospital with 1384 ward beds and 40 ED observation beds, with the approval of our institutional review board (IRB number: 202200931B0). We waived the need for consent from study participants because this was an observational study. Geriatric patients older than 65 years were eligible if they were suspected by ED physicians of having AUGIB on the basis of the presentation of hematemesis, melena, or coffee-ground vomiting. Patients who did not undergo endoscopic examination were excluded. 

### 2.2. Study Outcomes and Covariates

Triage nurses assessed the patients visiting the ED and recorded their initial vital signs. Patient clinical symptoms and biochemical examinations were obtained through chart review. We extracted patient data on underlying conditions, medical management, and clinical outcome (in-hospital mortality or survival to discharge) from electrical medical records. Physiological scoring systems were retrospectively calculated according to retrieved pertinent data. Shock index (SI) was calculated as heart rate (HR) divided by systolic blood pressure (SBP), and age SI was measured by multiplying SI and age [19,20]. The pre-endoscopy Rockall score, Rapid Acute Physiology Score (RAPS), Rapid Emergency Medicine Score (REMS), and Modified Early Warning Score (MEWS) are scoring systems with multiple components with weighted score points [21,22,23,24]; the components and details of each score are presented in Table 1, Table 2, Table 3 and Table 4. Patients who died during the hospitalization course were considered in-hospital mortality. In this study, we assessed the predictive performance of the pre-endoscopy Rockall score, SI, age SI, RAPS, REMS, and MEWS for in-hospital mortality.

### 2.3. Statistical Analysis

All statistical analyses were performed using SPSS statistics version 23.0 for Mac (IBM Corp., Chicago, IL, USA). The normality of the variables was tested using the Shapiro–Wilk test. Normally distributed variables were reported as mean ± standard deviation, and non-normally distributed variables were reported as median (minimum and maximum values). Categorical variables are presented as frequency and corresponding percentage (%). Fisher’s exact test and the Mann–Whitney *U* test were used to compare statistically significant differences between the mortality and non-mortality groups for categorical and numerical variables, respectively. Differences between the two groups are significant if *p* < 0.05. A subgroup analysis was performed for very elderly patients (older than 75 years) [25]. The mortality prediction of the six scoring systems in elderly patients with AUGIB was analyzed via univariate and multivariate logistic regressions. The variables which are significant in the univariate analysis (*p* < 0.05) were included in the multivariate regression model. The area under the receiver operating characteristic curve (AUROC) was calculated to evaluate the performance of the scoring systems in predicting mortality.

## 3. Results

### 3.1. Patient Characteristics

During the study period, 336 elderly AUGIB patients were enrolled; the mean age was 75 years (range from 65 to 94 years) and 69% of the patients were male. All enrolled patients were treated with intravenous proton pump inhibitor therapy and underwent endoscopic examination. The in-hospital mortality rate was 11.9%, and the mean age of those who died was 78 years old. Melena was more common in the non-mortality group, and coffee-ground vomitus was associated with poor prognosis; both are statistically significant (Table 5).

The mean ± SD of heart rate (HR), systolic blood pressure (SBP), and respiratory rate (RR) was 94.32 ± 18.70 per minute, 118.85 ± 33.21 mmHg, and 19.51 ± 2.64 per minute, respectively. The mortality group had a significantly lower SBP and Glasgow coma scale score than the survival group. In addition, the mortality group had a significantly higher HR and RR than the survival group (Table 5).

The biochemical examination showed that the mortality group had a higher white blood cell count, greater percentage of band cells, and longer prothrombin time (international normalized ratio) than the survival group. In addition, albumin levels were lower in the mortality group than in the non-mortality group (Table 5).

### 3.2. Performance of Clinical Scoring Systems

In the univariate logistic regression analysis, there was a significant difference between the mortality and non-mortality groups in pre-endoscopy Rockall score, SI, age SI, MEWS, REMS, and RAPS. The results of the multivariate logistic regression analysis show that higher MEWS, REMS, and RAPS were independent mortality predictors (Table 6). The AUROC was obtained to evaluate the prognostic value of each scoring system. While all six scoring systems were effective predictors of mortality, MEWS was found to be superior (AUC, 0.82; *p* < 0.01) (Figure 1).

### 3.3. Subgroup Analysis for Very Elderly Patients

A further subgroup analysis was performed for patients older than 75 years (defined as very elderly). There were 180 very elderly patients enrolled, of whom 27 died during the hospitalization course (mortality rate, 15%). In this subgroup, all scores except the pre-endoscopy Rockall score were significantly higher in the mortality group (Figure 2). MEWS, REMS, and RAPS better predict in-hospital mortality in very elderly patients than in elderly patients. MEWS also had the best predictive performance for in-hospital mortality in very elderly patients.

## 4. Discussion

To the best of our knowledge, this study is the first to evaluate MEWS, REMS, and RAPS for predicting in-hospital mortality of AUGIB in elderly and very elderly populations in the ED. We also compared these scoring systems with previously proposed clinical physiological scores, and the results show that MEWS had the best performance.

Endoscopy is an important modality for diagnosis and intervention for patients with AUGIB; however, the indication and timing of receiving endoscopic evaluation in the ED remain controversial [26,27]. Several national guidelines recommend the use of scoring systems to stratify patients with an elevated risk of mortality and for the potential benefits of early endoscopic evaluation in the ED. However, the most appropriate scoring system remains unclear [28].

Several scoring systems have been developed and validated to predict mortality due to AUGIB in the ED, and most of these scoring systems comprise demographic information (sex, age, comorbidities), vital signs, clinical symptoms, laboratory data, and endoscopic diagnosis [17,29]. Despite the completeness of these scoring systems, their complexity hinders their utility in a resource-limited or overcrowded ED [30]. Early and accurate assessments with an appropriate treatment plan are the cornerstone of managing AUGIB in the ED. Thus, simpler, bedside-available, and easily calculated scores have been proposed. Bozkurt et al. found that a MEWS score > 2 predicts in-hospital mortality in adult patients with AUGIB in the ED with an AUC of 0.722 [17]. Kocaoğlu et al. have proven that SI and age SI can quickly distinguish critical patients with gastrointestinal bleeding in the ED [18].

In 2001, Subbe CP et al. first introduced MEWS as a scoring system to predict the risk of developing catastrophic medical events [24]. MEWS is repeatable, can be rapidly calculated, is bedside-available, and is widely used in the ED. As MEWS includes cardiac and pulmonary parameters, such as HR, RR, and blood pressure, which sensitively reflect patients’ circulation and fluid status, MEWS is able to assist in the earlier detection of patients who are becoming unstable and guides important interventions. Studies have demonstrated the reliable performance of MEWS for risk stratification in sepsis, trauma, and critical patients with ICU readmission [31,32,33]. In elderly patients presenting to the ED, MEWS has also been proven to be an effective tool for predicting in-hospital mortality [34,35]. Dundar et al. reported that for elderly patients older than 65 years, hospitalization and mortality were associated with MEWS in the ED [35]. Moreover, MEWS has excellent accuracy in predicting mortality in elderly patients with COVID-19 and community-acquired pneumonia [36,37]. However, the performance of MEWS in predicting mortality in elderly patients with AUGIB in the ED was unknown.

Our study reveals that MEWS, REMS, and RAPS are independent predictors of mortality in elderly AUGIB patients, with MEWS having the best performance. The target population may be the reason for the superiority of MEWS to the other five scores. Anti-hypertensive and anti-arrhythmic medications are commonly used in the geriatric population, resulting in a blunted change in HR at the early stage of hypovolemic shock and further modulating SI. Furthermore, the geriatric population exhibits the effects of age, which is the weighted parameter in age SI, pre-endoscopy Rockall score, and REMS but not in MEWS. Moreover, massive blood loss is a common cause of AUGIB-related death which may present as the lethal triad (coagulopathy, metabolic acidosis, and hypothermia). Unlike the other five scores, MEWS is the only score that assesses the patient’s body temperature, which may explain the better predictive capability in this population. Furthermore, hemodynamic change due to blood loss is not as sensitive in a geriatric population as in a younger population. This is because elderly patients have a higher baseline blood pressure and might not present with compensatory tachycardia because of their blunted response and polypharmacy status, making the early assessment of blood loss challenging [38].

We further analyzed a subgroup of very elderly patients: MEWS, REMS, and RAPS showed better mortality predictive performance in this population than in the elderly population. This might have resulted from the link between chronic cognitive impairment and increased comorbidities, which is a major factor contributing to poor outcomes in geriatric patients with AUGIB [39]. Furthermore, compared with younger patients, elderly patients with acute illness tend to present with cognitive dysfunction or delirium [40]. Acute mental status changes and chronic cognitive impairment frequently coexist in the geriatric population, and both are poor prognosis indicators [41]. Therefore, clinical physiological scoring systems that include assessments of mental status changes can sensitively reflect disease severity, which explains the superiority of MEWS, REMS, and RAPS in very elderly populations.

Clinical physiological scoring systems can help physicians rapidly stratify patients according to risk and quickly detect those who are becoming unstable, even without laboratory or endoscopic results. In the present study, we demonstrated that MEWS is superior to pre-endoscopy Rockall score, SI, age SI, RAPS, and REMS in predicting in-hospital mortality among elderly and very elderly patients with AUGIB in the ED.

## 5. Limitations

This study has some limitations. First, this was a retrospective, single-hospital study, and some data were missing. Prospective multicenter studies are warranted for further validation. Second, coexisting conditions, such as infectious or cardiovascular diseases, were not investigated, which may have affected the score without being directly related to AUGIB. Third, AUGIB has different etiologies, including peptic ulcer, varices, Mallory–Weiss syndrome, malignancy, etc., which have different presentations and clinical outcomes; therefore, further studies conducted according to specific etiology are needed. Fourth, the prevalence and etiologies of AUGIB are different among various regions and populations [5,7]. Because all our enrolled patients are Asian, additional research is necessary to compare these findings across different ethnic groups.

## 6. Conclusions

The present study compares six clinical physiological scoring systems, including pre-endoscopy Rockall score, SI, age SI, RAPS, REMS, and MEWS, to predict in-hospital mortality in elderly and very elderly patients with AUGIB in the ED, with MEWS having the best discrimination (AUROC, 0.82). This simple and reliable tool may facilitate risk stratification and decision making in this vulnerable group. Further etiology-specific multicenter prospective studies for various ethnic groups are warranted for further investigation.

## Figures and Tables

**Figure 1 medicina-59-00556-f001:**
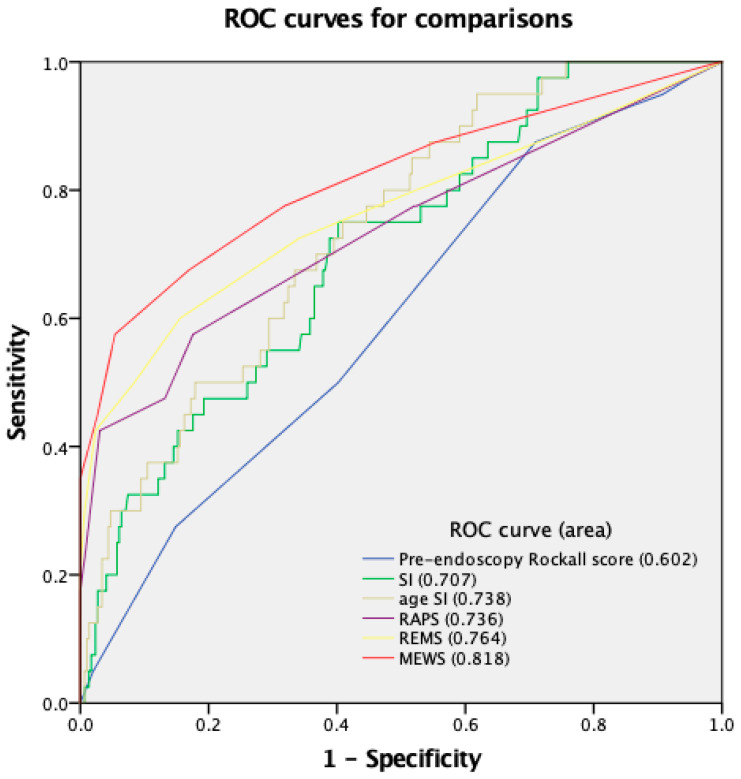
ROC curves for six scoring systems in evaluation of mortality for patients older than 65 years.

**Figure 2 medicina-59-00556-f002:**
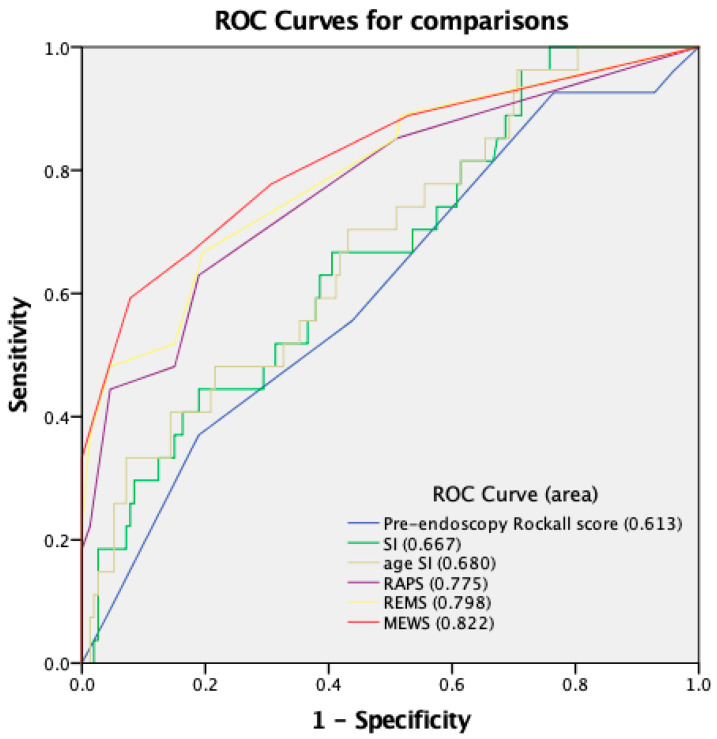
ROC curves for six scoring systems in evaluation of mortality for patients older than 75 years.

**Table 1 medicina-59-00556-t001:** Pre-endoscopy Rockall score [21].

Score				
Variable	0	+1	+2	+3
Age (years)	<60	60–70	≥80	
Shock				
HR (/min)	<100	>100	>100	
SBP (mmHg)	>100	>100	<100	
Comorbidity	None		IHD, CHF, any major comorbidity	Renal/liver failure, metastatic malignancy

HR: heart rate, SBP: systolic blood pressure, IHD: ischemic heart disease, CHF: congestive heart failure.

**Table 2 medicina-59-00556-t002:** Rapid Acute Physiology Score (RAPS) [22].

Score					
Variable	0	+1	+2	+3	+4
HR (/min)	70–109		55–69 110–139	40–54 140–179	≤39 ≥180
MAP (mmHg)	70–109		55–69 110–139	130–159	≤49 ≥160
RR (/min)	12–24	10–11 25–34	6–9	35–49	≤5 ≥50
GCS	≥14	11–13	8–10	5–7	≤4

HR: heart rate, MAP: mean arterial pressure, RR: respiratory rate, GCS: Glasgow coma scale.

**Table 3 medicina-59-00556-t003:** Rapid Emergency Medicine Score (REMS) [23].

Score							
Variable	0	+1	+2	+3	+4	+5	+6
Age (years)	<45		45–54	55–64		65–74	≥74
HR (/min)	70–109		55–69 110–139	40–54 140–179	≤39 >179		
MAP (mmHg)	70–109		55–69 110–129	130–159	≤49 >159		
RR (/min)	12–24	10–11 25–34	6–9	35–49	≤5 >49		
GCS	≥14	11–13	8–10	5–7	3 or 4		
SpO_2(_%)	>89	86–89		75–85	<75		

HR: heart rate, MAP: mean arterial pressure, RR: respiratory rate, GCS: Glasgow coma scale.

**Table 4 medicina-59-00556-t004:** Modified Early Warning Score (MEWS) [24].

Score				
Variable	0	+1	+2	+3
SBP (mmHg)	101–119	81–100	71–80 ≥200	<70
HR (/min)	51–100	41–50 101–110	<40 111–129	≥130
RR (/min)	9–14	15–20	<9 21–29	≥30
BT (°C)	35–38.4		<35 ≥38.5	
AVPU score	Alert	Reacts to Voice	Reacts to Pain	Unresponsive

SBP: systolic blood pressure, HR: heart rate, RR: respiratory rate, BT: body temperature.

**Table 5 medicina-59-00556-t005:** Comparisons of baseline characteristics of the patients according to in-hospital mortality.

Variable	Mortality Group (*n* = 40)	Non-Mortality Group (*n* = 296)	*p* Value
Age ^b^	78 (73–85)	75 (68–79)	<0.01 *
Sex, M/F	27/13 (2.08:1)	205/91 (2.25:1)	0.82
Vital signs			
BT (°C) ^b^	36.0 (35.6–37.0)	36.2 (35.8–36.7)	0.61
Heart rate (/min) ^a^	100.70 ± 20.49	93.45 ± 18.31	0.02 *
SBP (mmHg) ^a^	101.70 ± 34.32	121.17 ± 32.43	<0.01 *
DBP (mmHg) ^b^	63 (50–77)	69 (58–83)	0.08
MAP (mmHg) ^b^	75 (57–93)	85 (72–103)	<0.01 *
Respiratory rate (/min) ^a^	22.58 ± 5.43	19.10 ± 1.60	<0.01 *
GCS ^b^	12 (7–15)	15 (14–15)	<0.01 *
Symptom, *n* (%)			
Hematemesis	10 (25.0)	46 (15.5)	0.13
Coffee-ground vomiting	14 (35.0)	36 (12.2)	<0.01 *
Melena	16 (40)	189 (63.9)	<0.01 *
Others	0 (0)	33 (11.1)	0.03 *
Lab test results			
Hemoglobin (g/dL) ^a^	8.55 ± 2.68	9.30 ± 2.74	0.10
Platelets (×10^3^/μL) ^b^	155 (93–297)	188 (126–248)	0.95
WBC (×10^3^/μL) ^b^	11.9 (6.6–18.4)	8.5 (6.2–11.6)	<0.01 *
Band (%) ^b^	0.25 (0–4.2)	0 (0–0)	<0.01 *
Albumin ^a^	2.85 ± 0.63	3.49 ± 0.59	<0.01 *
Bilirubin ^b^	1.1 (0.6–2.0)	1.0 (0.7–1.4)	0.33
BUN ^b^	42.2 (27.4–67.2)	36.7 (21.4–56.1)	0.21
Creatinine (mg/dL) ^b^	1.5 (0.9–2.5)	1.2 (0.9–1.8)	0.97
INR ^b^	1.2 (1.1–1.4)	1.1 (1.0–1.2)	0.05 *
Comorbidity, *n* (%)			
Hypertension	25 (62.5)	169 (57.1)	0.52
DM	15 (37.5)	105 (35.5)	0.80
CVD	10 (25)	39 (13.3)	0.05 *
CVA	6 (15)	6 (2)	<0.01*
CKD	4 (10)	31 (10.5)	0.93
COPD	3 (7.5)	5 (1.7)	0.02 *
Cirrhosis	12 (30)	50 (16.9)	0.05 *
Malignancy	17 (42.5)	36 (12.2)	<0.01 *

Data presented as ^a^ mean ± standard deviation, ^b^ median (minimum–maximum), or *n* (%) values. BT: body temperature, SBP: systolic blood pressure, DBP: diastolic blood pressure, MAP: mean arterial pressure, GCS: Glasgow coma scale, DM: diabetes mellitus, CVD: cardiovascular disease, CVA: cerebrovascular accident, CKD: chronic kidney disease, COPD: chronic obstructive pulmonary disease. * means *p* Value < 0.05.

**Table 6 medicina-59-00556-t006:** Comparison of six scoring systems between the two groups.

Scoring System	Mortality Group (n = 40)	Non-Mortality Group (n = 296)	Univariate		Multivariate	
			OR (95%CI)	*p* Value	OR (95%CI)	*p* Value
Pre-endoscopy Rockall	4.62 ± 1.27	4.14 ± 1.32	1.34 (1.03–1.76)	0.03 *	1.24 (0.72–2.14)	0.44
SI	1.08 ± 0.36	0.84 ± 0.31	6.83 (2.75–16.94)	<0.01 *	2.18 (0.39–12.37)	0.38
Age SI	85.33 ± 30.49	62.41 ± 24.59	1.03 (1.02–1.04)	<0.01 *	1.02 (0.98–11.81)	0.17
RAPS	2.65 ± 2.88	1.39 ± 1.52	1.72 (1.44–2.06)	<0.01 *	1.61 (1.14–2.28)	<0.01 *
REMS	9.90 ± 3.49	6.91 ± 1.61	1.74 (1.46–2.06)	<0.01 *	1.72 (1.20–2.47)	<0.01 *
MEWS	5.00 ± 2.61	2.12 ± 1.32	2.16 (1.75–2.66)	<0.01 *	2.14 (1.39–3.31)	<0.01 *

* means *p* Value <0.05.

## Data Availability

The datasets generated and/or analyzed during the current study are available from the corresponding author upon reasonable request.

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
