# Peer review of "Performance of Six Clinical Physiological Scoring Systems in Predicting In-Hospital Mortality in Elderly and Very Elderly Patients with Acute Upper Gastrointestinal Bleeding in Emergency Department"

_medicina, 2023, doi:10.3390/medicina59030556_

Round 1

Reviewer 1 Report

Thank you for the interesting paper.

I have some minor comments:

Introduction

Although your study is originating from Asia, you cited most papers from north American population especially in the introduction. i recommend use more international data and specifically comment in the situation in your region or country. You can also make a comparison as it would be interesting to see if there is a significant difference. 

Results 

Section 3.2 why you only performed univariate logistic regression and not multivariate? It should be valuable to see if these scores are still a significant predictors of mortality. 

References

I think the number is too high and you can use around half of these References and still convey the same information

Author Response

Reviewer # 1

  1. Although your study is originating from Asia, you cited most papers from north American population especially in the introduction. I recommend use more international data and specifically comment in the situation in your region or country. You can also make a comparison as it would be interesting to see if there is a significant difference. 

Author’s response: 

We concur with the reviewer's opinion that the incidence and etiology of acute upper gastrointestinal bleeding (AUGIB) can vary among different populations and geographic regions. Accordingly, utilizing international data would be more suitable than solely relying on North American data. To provide a more comprehensive understanding, we have included references from Asia in the introduction sections. (line 43-44)

Given that our study exclusively involved Asian participants, it is crucial to note that further research is necessary to compare these findings with those of other ethnic groups. Therefore, we have acknowledged this as a limitation of our study. (line 253-255; line 261-263)

Line 43-44:

“The studies conducted on Asian populations also revealed a similar mortality rate range [5, 6].”

Line 253-255:

“Fourth, the prevalence and etiologies of AUGIB are different among various regions and populations [5, 7]. Since all of our enrolled patients are Asians, additional research is necessary to compare these findings across different ethnic groups.”

Line 261-263:

“Further etiology-specific multicenter prospective studies for various ethnic groups are warranted for further investigation.”

  1. Section 3.2 why you only performed univariate logistic regression and not multivariate? It should be valuable to see if these scores are still significant predictors of mortality.

Authors’ response:

We greatly appreciate the valuable feedback provided by the reviewer. As a result, we conducted multivariate logistic regression analysis to account for potential confounding effects. Additionally, we revised Table 6 by integrating the findings of both univariate and multivariate logistic regression analyses, leading to the omission of the first paragraph in section 3.2 that initially reported the results of the univariate logistic regression analysis. We also made revisions to the method, result, and discussion paragraphs. (line 110-113; line 149-153; line 212-214)

Line 110-113:

“The mortality prediction of the six scoring systems in elderly patients with AUGIB was analyzed via univariate and multivariate logistic regression. The variables which were significant in univariate analysis (P < 0.05) were included in multivariate regression model.”

Line 149-153:

“In the univariate logistic regression analysis, there was a significant difference between the mortality and non-mortality groups in pre-endoscopy Rockall score, SI, age SI, MEWS, REMS and RAPS. Results of the multivariate logistic regression analysis showed that higher MEWS, REMS, and RAPS were independent mortality predictors (Table 6).”

Line 212-214:

“Our study revealed MEWS, REMS, and RAPS were independent predictors for mortality in elderly AUGIB patients, with MEWS having the best performance.”

To ensure clarity, we have included both the original and revised versions of Table 6 below.

Original

Table 6. Comparison of six scoring systems between the two group

Scoring system

Mortality

group(n=40)

Non-mortality group(n=296)

P value

Pre-endoscopy Rockall

4.62 ± 1.27

4.14 ± 1.32

0.03*

SI

1.08 ± 0.36

0.84 ± 0.31

<0.01*

Age SI

85.33 ± 30.49

62.41 ± 24.59

<0.01*

RAPS

2.65 ± 2.88

1.39 ± 1.52

<0.01*

REMS

9.90 ± 3.49

6.91 ± 1.61

<0.01*

MEWS

5.00 ± 2.61

2.12 ± 1.32

<0.01*

Revised

Table 6. Comparison of six scoring systems between the two group

Scoring system

Mortality

group(n=40)

Non-mortality group(n=296)

Univariate

Multivariate

OR (95%CI)

P value

OR (95%CI)

P value

Pre-endoscopy Rockall

4.62 ± 1.27

4.14 ± 1.32

1.34(1.03–1.76)

0.03*

1.24(0.72-2.14)

0.44

SI

1.08 ± 0.36

0.84 ± 0.31

6.83(2.75–16.94)

<0.01*

2.18(0.39-12.37)

0.38

Age SI

85.33 ± 30.49

62.41 ± 24.59

1.03(1.02–1.04)

<0.01*

1.02(0.98-11.81)

0.17

RAPS

2.65 ± 2.88

1.39 ± 1.52

1.72(1.44–2.06)

<0.01*

1.61(1.14-2.28)

<0.01*

REMS

9.90 ± 3.49

6.91 ± 1.61

1.74(1.46–2.06)

<0.01*

1.72(1.20-2.47)

<0.01*

MEWS

5.00 ± 2.61

2.12 ± 1.32

2.16(1.75–2.66)

<0.01*

2.14(1.39-3.31)

<0.01*

  1. I think the number is too high and you can use around half of these References and still convey the same information

Authors’ response:

In response to reviewer’s suggestion, we simplified the references and revised the manuscript accordingly.

Reviewer 2 Report

The paper is an interesting report, well written, and it is presented in a proper form. 

I suggest to check the English consistency of the manuscript. There are several ortographical errors throughout the manuscript.

Author Response

Reviewer # 2

  1. I suggest to check the English consistency of the manuscript. There are several ortographical errors throughout the manuscript.

Author’s response: 

We greatly appreciate your attention to detail and the valuable feedback. We have carefully reviewed our manuscript and taken note of the orthographical errors you have pointed out. Furthermore, to ensure the consistency of English, we have submitted the manuscript for editing by a professional English editor and provided the English editing certification as shown below. We hope that the revised manuscript meets your expectations.